# Pseudo Meets Zero: Boosting Zero-Shot Composed Image Retrieval with Synthetic Images

## Abstract

Composed Image Retrieval (CIR) employs a triplet architecture to combine a reference image with modified text for target image retrieval. To mitigate high annotation costs, Zero-Shot CIR (ZS-CIR) methods eliminate the need for manually annotated triplets. Current methods typically map images to tokens and concatenate them with modified text. However, they encounter challenges during inference, especially with fine-grained and multi-attribute modifications. We argue that these challenges stem from insufficient explicit modeling of triplet relationships, which complicates fine-grained interactions and directional guidance. To this end, we propose a Synthetic Image-Oriented training paradigm that automates pseudo target image generation, facilitating efficient triplet construction and accommodating inherent target ambiguity. Furthermore, we propose the Pseudo domAiN Decoupling-Alignment (**PANDA**) model to mitigate the *Autophagy* phenomenon caused by fitting targets with pseudo images. We observe that synthetic images are intermediate between visual and textual domains in triplets. Regarding this phenomenon, we design the Orthogonal Semantic Decoupling module to disentangle the pseudo domain into visual and textual components. Additionally, Shared Domain Interaction and Mutual Shift Constraint modules are proposed to collaboratively constrain the disentangled components, bridging the gap between pseudo and real triplets while enhancing their semantic consistency. Extensive experiments demonstrate that PANDA outperforms existing state-of-the-art methods across two general scenarios and two domain-specific CIR datasets.

## 1 Introduction

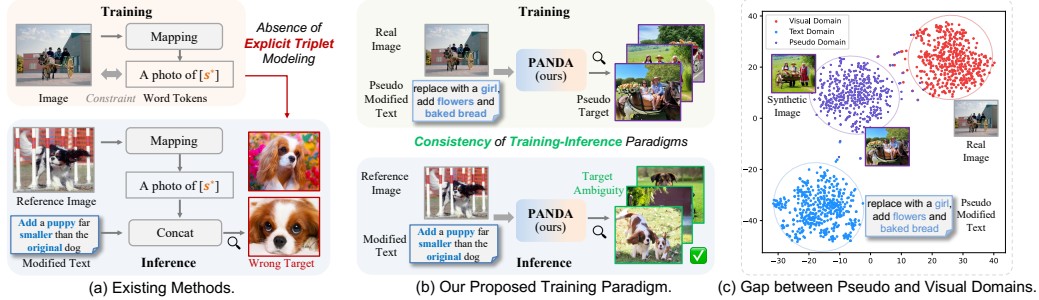

(a) Existing Methods.  (b) Our Proposed Training Paradigm.  (c) Gap between Pseudo and Visual Domains.

Figure 1: Illustrations of the motivation for our training paradigm and approach: (a) Existing ZS-CIR paradigm. (b) Our proposed training paradigm. (c) We observe a domain gap between synthetic and real images within (b), where reducing this gap aids in further unifying training and inference.

Composed Image Retrieval (CIR) retrieves a target image by integrating a reference image and modified text, achieving notable advancements recently Vo et al. (2019); Delmas et al. (2022); Yang et al. (2024b). However, constructing reference-modified text-target triplets, particularly in domain-specific contexts like e-commerce, is resource-intensive Karthik et al. (2024b;a). Consequently, Zero-Shot Composed Image Retrieval (ZS-CIR) has emerged, focusing on scenarios that eliminate the need for manually annotated triplets Baldrati et al. (2023); Lin et al. (2024). The current ZS-CIR

framework trains on image-caption datasets to map images to tokens for the text encoder Tang et al. (2024); Du et al. (2024). During inference, the reference image is tokenized and concatenated with modified text, allowing the text encoder to extract features for target image retrieval.

However, the existing framework struggles with fine-grained or multi-attribute modifications due to the lack of explicit triplet modeling. It overlooks two key roles of the modified text in CIR: *Interacting with the reference.* Current methods implicitly assign the crucial interaction between the modified text and reference image to the text encoder Saito et al. (2023); Baldrati et al. (2023), missing essential contextual cues from complex visual-textual semantics. *Guiding from reference to target.* The modified text should guide retrieval by outlining the differences between reference and target images Kim et al. (2021). Existing methods mistakenly treat it as a direct descriptor of target features, hindering effectiveness in scenarios involving multiple attributes or new elements.

To address the aforementioned issues, we propose a Synthetic Image-Oriented (SIO) training paradigm tailored for the ZS-CIR task. This approach aims to automate the construction of pseudo target images using generative models, thereby creating triplets similar to those encountered during inference. This training paradigm presents several advantages: *(i) Target Ambiguity Alignment.* The ZS-CIR task inherently involves target ambiguity Liu et al. (2021); Delmas et al. (2022), allowing multiple valid options to fulfill the modified text's objectives. Diffusion generative models can produce multiple images based on the same semantics Croitoru et al. (2023); Wu et al. (2023), making them well-suited for this characteristic. *(ii) Efficient Triplet Construction.* This paradigm utilizes existing image-caption datasets to rapidly generate pseudo triplets. Current ZS-CIR methods implicitly learn semantic correspondences within triplets Saito et al. (2023); Tang et al. (2024), often depending on large datasets (e.g., CC3M Sharma et al. (2018)). In contrast, SIO requires significantly less data, up to two orders of magnitude less than existing ZS-CIR methods.

Nevertheless, recent studies Alemohammad et al. (2024) indicate that merely using synthetic images for training can lead to performance degradation due to the **Autophagy** phenomenon. We observe that synthetic images exist in an intermediate pseudo domain between the visual and textual domains, as shown in Figure 1 (c). To prevent the model from converging excessively to the pseudo domain and to enhance the performance gains of pseudo triplets, we propose the Pseudo Domain Decoupling-Alignment (**PANDA**) model. We first introduce an Orthogonal Semantic Decoupling (OSD) module, which explicitly disentangles the features of the pseudo domain into two complementary parts. The first part focuses on aligning with the visual domain of the target image in the actual triplet, while the second part emphasizes constraints with the textual domain of the modified text. For the first part, we propose a Shared Domain Interaction (SDI) module that employs shared network weights and specific learnable tokens to model interactions among the multimodal, visual, textual, and pseudo domains. By progressively interacting the real image side and the modified text within the pseudo triplets, a multimodal representation that fully integrates both components is obtained. For the second part, we design a Mutual Shift Constraint (MSC) module that captures the differences from the reference to the target, constrained by the modified text.

In a nutshell, our contributions are summarized as follows:

- A new training paradigm is proposed to automate pseudo target image generation, facilitating efficient triplet construction and addressing target ambiguity in the ZS-CIR task.

- We gain insight into the fact that synthetic images exist in an intermediate state between visual and textual domains, underscoring the need for specialized modeling.

- The proposed PANDA model focuses on mitigating the Autophagy phenomenon when using pseudo images as targets while enhancing semantic interactions and alignment among triplets.

- Extensive experiments show that PANDA outperforms state-of-the-art methods across two general scenarios and two domain-specific datasets.

## 2 RELATED WORK

### 2.1 ZERO-SHOT COMPOSED IMAGE RETRIEVAL (ZS-CIR).

Currently, methods in the field of CIR can be broadly categorized into two paradigms. The first paradigm employs fully supervised *late fusion* methods Baldrati et al. (2022); Chen et al. (2024b);

Jiang et al. (2024a); Yang et al. (2024b), using manually annotated reference-modified text-target triplets as training data Liu et al. (2024a); Han et al. (2023); Zhang et al. (2024). For instance, Baldrati et al. (2022) proposes a simple yet effective fusion model, Combiner, to combine features extracted by the CLIP Radford et al. (2021) model. The second paradigm, commonly used in ZS-CIR settings, adopts the *word token* framework Tang et al. (2024); Bai et al. (2024), initially introduced by Saito et al. (2023). This method learns to map image embeddings into word tokens interpretable by a text encoder during the training phase on image-caption pairs. In the testing phase, the modified text is concatenated directly for target retrieval. Lin et al. (2024) further enhances the fine-grained representation of reference images by mapping them into subject-oriented word tokens and several attribute-oriented word tokens. However, current methods overlook explicit modeling of triplet semantics, neglecting the core functions of modified text to interact with the reference and guide it toward the target. Our approach addresses these issues at both the training paradigm and methodological levels, achieving enhanced semantic interaction and alignment among triplets.

### 2.2 DATA AUGMENTATION USING SYNTHETIC IMAGES.

With the rapid advancement of text-to-image generation models Dhariwal & Nichol (2021); Li et al. (2019); Ding et al. (2021); Li et al. (2023b), an increasing number of pioneering works are applying synthetic data to computer vision and multimodal tasks Wang et al. (2021); Wood et al. (2021); Yang et al. (2024a). In domains where labeled data is costly, such as medical applications, synthetic images can mitigate data scarcity, facilitating model learning Chen et al. (2021); Usman Akbar et al. (2024); Müller-Franzes et al. (2023). In general image tasks like classification and segmentation, synthetic images serve as excellent data augmentation for real-world images and can be used as the entire training dataset due to their high-quality generation. Tian et al. (2024); Fan et al. (2024); Liu et al. (2024b). He et al. (2023) use high-quality synthetic images, filtering out low-quality samples, and achieve significant improvements over the CLIP model. Hammoud et al. (2024) propose training CLIP models Radford et al. (2021) solely with synthetic text-image pairs generated by text-to-image models and large language models. This scalable method eliminates manual intervention and matches the performance of CLIP models trained on real data. Inspired by pioneering studies, we introduce synthetic images into the ZS-CIR task for two key reasons: (i) Diffusion models can generate multiple images from the same semantics Croitoru et al. (2023); Wu et al. (2023), aligning with the inherent target ambiguity in CIR Liu et al. (2021); Delmas et al. (2022). (ii) Efficiently constructs pseudo triplets from existing image-caption datasets.

## 3 APPROACH

In the following sections, we will present the problem formulation of ZS-CIR in Section 3.1, introduce the SIO training paradigm in Section 3.2, provide insights on optimizing target retrieval using multiple synthetic images in Section 3.3, detail the PANDA model in Section 3.4, and outline the training and inference processes in Section 3.5.

### 3.1 PROBLEM FORMULATION.

The objective of ZS-CIR is to learn how to retrieve a target image $I_{\text{tar}}$ at inference time without providing manually annotated reference-modifier-target triplets, utilizing a reference image $I_{\text{ref}}$ and user-provided modified text $t_{\text{mod}}$. We propose a novel Synthetic Image-Oriented training paradigm. Given an image-caption dataset $D_{IC} = \{(I_i, C_i)\}_{i=1}^{N}$, we construct pseudo-triplets $\{(I_{\text{ref}}, T_{\text{mod}}, I_{\text{tar}})\}$ using an image generation model. Our objective is to perform associative learning using the embeddings extracted from $I_{\text{ref}}$, $t_{\text{mod}}$, and $I_{\text{tar}}$, respectively, in order to learn a mapping function $f : (I_{\text{ref}}, T_{\text{mod}}) \rightarrow I_{\text{tar}}$. The learned function $f$ is then evaluated on real triplets $\{(I_{\text{ref}}^*, T_{\text{mod}}^*, I_{\text{tar}}^*)\}$ to assess performance in retrieval tasks.

### 3.2 SYNTHETIC IMAGE-ORIENTED (SIO) TRAINING.

We propose a Synthetic Image-Oriented training paradigm, which automates the construction of pseudo-triplets $\{(I_{\text{ref}}, T_{\text{mod}}, I_{\text{tar}})\}$ following two strategies: *Fine-grained* and *Coarse-grained*.

**Fine-grained.** This strategy focuses on modifying local objects within the image, achieving fine-grained semantics in the modified text. From the dataset $D_{IC}$, we select a subset $D'_{IC}$. For a given image-caption pair $I_i, C_i$, we use the following prompt to guide an LLM in generating new captions and adding or replacing objects within the origin image. The prompt is designed as: "*You are a painter. Given a caption $[C_i]$, carefully add or replace reasonable and simple objects to the caption for the painting, answer with three short phrases: 1. New caption: 2. New added objects: 3. New replaced objects: Answer:*". We denote the new caption generated by the LLM as $C'_i$, the added objects as $T_{add}$ and the replaced objects as $T_{rep}$. Using $C'_i$, we generate a batch of $N_{\text{gen}}$ images $\{I_i^{\text{gen}}\}$ through an image generation model. Subsequently, we create the modified text $T_{\text{mod}}^{\text{fine}}$ based on $C'_i$, $T_{add}$ and $T_{rep}$ by following predefined templates, such as: "*add $[T_{add}]$ and change to $[C'_i]$*", or "*replace to $[T_{rep}]$*". This results in pseudo-triplets $(I_i, T_{rep}^{\text{fine}}, \{I_i^{\text{gen}}\})$.

**Coarse-grained.** This strategy emphasizes global image semantic replacement, resulting in coarse-grained semantics. We select an image $I_a$ from the subset dataset $D'_{IC}$ along with its most similar image $I_b$ based on the embeddings extracted from the CLIP model, where $I_a$ serves as the reference image and $I_b$ as the target image. Next, for their respective captions $C_a$ and $C_b$, we generate the modified text using a template, such as: "*change $[C_a]$ to $[C_b]$.*" The strategy results in pseudo-triplets $(I_a, T_{\text{mod}}^{\text{coarse}}, I_b)$, which simulate the transformation from one image to another, addressing cases where object replacement is involved. To make full use of the target ambiguity across multiple synthetic images, we assign a weight $w$ to balance the contributions of the two strategies, ensuring $w_{\text{fine}} > w_{\text{coarse}}$ to alleviate the inherent target ambiguity in the CIR task and to strengthen fine-grained associations within the triplet.

To clarify the model architecture, we will refer to the elements in the constructed $(I_i, T_{\text{mod}}^{\text{fine}}, \{I_i^{\text{gen}}\})$ and $(I_a, T_{\text{mod}}^{\text{coarse}}, I_b)$ as the reference image, modified text, and target image in the following sections.

### 3.3 THEORETICAL INSIGHTS

In this section, we justify the rationale behind introducing pseudo-triplets, each containing multiple synthetic images as possible targets, and explain how our train pattern outperforms existing solutions. In the ZS-CIR task, we hypothesize that there is an underlying ground truth mapping function $\mathcal{F}$ and aim to get an approximated function $f$ that correctly retrieves the targets in available triplets for training. Therefore, we theoretically construct a toy problem for the ZS-CIR task.

**Toy Problem.** *Given a triplet $S_{tri} = \{(I_{ref}, T_{mod}, I_{tar})\}$, the objective is to learn an approximated mapping function $f$ that satisfies $f((I_{ref}, T_{mod}, I_{tar})) - \mathcal{F}((I_{ref}, T_{mod}, I_{tar})) = 0$. In another word, the composed function $f - \mathcal{F}$ takes $S_{tri}$ as its root.*

Existing ZS-CIR methods, where only one target image is paired with a reference and modified text, define a triplet $(I_{\text{ref}}, T_{\text{mod}}, I_{\text{tar}})$ that results in a linear approximation relative to $f - \mathcal{F}$. In contrast, our approach introduces multiple synthetic target images in a pseudo-triplet $(I_i, T_{\text{mod}}^{\text{fine}}, \{I_i^{\text{gen}}\})$, where each target acts as a root for $f = \mathcal{F}$, facilitating polynomial approximations.

**Weierstrass Approximation Theorem.** *Let $\mathcal{F}$ be a continuous real-valued function on the interval $[a, b]$. For any $\epsilon > 0$, there exists a polynomial $f$ such that for all $x \in [a, b]$, $|f(x) - \mathcal{F}(x)| < \epsilon$. Furthermore, the approximation error can be bounded as follows: if $\mathcal{F}$ has a continuous $k$-th derivative, then for any $n \in \mathbb{N}$, there exists a polynomial $f_n$ of degree at most $n$ such that:*

$$|f_n(x) - \mathcal{F}(x)| \leq \frac{\pi}{2} \frac{1}{(n+1)^k} |\mathcal{F}^{(k)}| \tag{1}$$

Moreover, leveraging our toy problem definition and the Weierstrass approximation theorem, multiple targets allow for higher-degree polynomial functions, resulting in more accurate approximations and reduced error bounds. This offers theoretical support for the effectiveness of our approach.

### 3.4 PSEUDO DOMAIN DECOUPLING-ALIGNMENT (PANDA)

**Shared Domain Interaction (SDI).** The SDI module leverages shared model parameters to simultaneously model multimodal, visual (also pseudo), and textual inputs. First, for processing multimodal inputs in the **SDI I** architecture, we enhance fine-grained interactions between the modified

Figure 2: Illustration of the training and inference process of the proposed PANDA, along with the SDI, OSD, and MSR modules. OSD: Decouples the pseudo domain into visual domain semantics $\mathbf{z}_{\mathcal{P}}^{\mathcal{V}}$ and textual domain semantics $\mathbf{z}_{\mathcal{P}}^{\mathcal{T}}$, constraining visual domain semantics through the triplet retrieval process. SDI: Three setups (I-III) handle multimodal, visual, and textual inputs, respectively. MSR: Constrains textual domain semantics $\mathbf{z}_{\mathcal{P}}^{\mathcal{T}}$ through mutual shift semantic modeling.

text and image by extracting image patch features $\mathbf{v}_{\mathrm{ref}} \in \mathbb{R}^{m \times D}$ (with $m$ being the patch number and $D$ being the embedding dimension) output from the second-to-last layer of the frozen CLIP visual encoder. Next, for multimodal inputs, we define a set of $n$ learnable tokens $\mathbf{z}_{\mathrm{i}} \in \mathbb{R}^{n \times D}$ (where $n$ being the number of learnable tokens) and employ an off-the-shelf Transformer network, which comprehensively models interactions via the multi-head self-attention (SA) and cross-attention (CA) mechanisms. We adopt a progressive strategy where the reference & modified side multimodal learnable tokens $\mathbf{z}_{\mathcal{M}}$ first interact with the tokenized modified text embeddings $\mathbf{t}_{\mathrm{mod}}$ through the SA layers to capture textual semantic representations, followed by further semantic interaction with $\mathbf{v}_{\mathrm{ref}}$ in the CA layer. This process is formulated as follows:

$$\mathbf{z}_{\mathcal{M}} = \mathrm{FC}_{\mathcal{M}}(\mathcal{F}_{\mathrm{CA}}(\mathcal{F}_{\mathrm{SA}}([\mathbf{z}_{\mathrm{i}}; \mathbf{t}_{\mathrm{mod}}]), \mathbf{v}_{\mathrm{ref}})) \tag{2}$$

where $[x; y]$ denotes represents the concatenation of embeddings $x$ and $y$. For the **SDI II** architecture designed for visual inputs, a similar approach is employed. The learnable tokens $\mathbf{z}_{\mathrm{ii}}$ interact with $\mathbf{v}_{\mathrm{ref}}$ in the CA layer, yielding visual domain representation tokens $\mathbf{z}_{\mathcal{V}}$. In the case of the **SDI III**, which addresses textual inputs, the learnable tokens $\mathbf{z}_{\mathrm{iii}}$ engage with $\mathbf{t}_{\mathrm{mod}}$ in the SA layer, resulting in textual domain representation tokens $\mathbf{z}_{\mathcal{T}}$. This interaction can be formulated as follows:

$$\mathbf{z}_{\mathcal{V}} = \mathrm{FC}_{\mathcal{V}}(\mathcal{F}_{\mathrm{CA}}(\mathcal{F}_{\mathrm{SA}}(\mathbf{z}_{\mathrm{ii}}), \mathbf{v}_{\mathrm{ref}})), \quad \mathbf{z}_{\mathcal{T}} = \mathrm{FC}_{\mathcal{T}}(\mathcal{F}_{\mathrm{CA}}(\mathcal{F}_{\mathrm{SA}}([\mathbf{z}_{\mathrm{iii}}; \mathbf{t}_{\mathrm{mod}}]))) \tag{3}$$

**Orthogonal Semantics Decoupling (OSD).** For the synthetic image $I_{\mathrm{tar}}$, the OSD module facilitates decoupling to mitigate over-fitting to the pseudo domain. To differentiate it from the vision domain of real images, $I_{\mathrm{tar}}$ is represented using the tokens $\mathbf{z}_{\mathcal{P}}$ derived from SDI II as follows:

$$\mathbf{z}_{\mathcal{P}} = \mathcal{F}_{\mathrm{CA}}(\mathcal{F}_{\mathrm{SA}}(\mathbf{z}_{\mathrm{ii}}), \mathbf{v}_{\mathrm{tar}}) \tag{4}$$

where $\mathbf{v}_{\mathrm{tar}} \in \mathbb{R}^{m \times D}$ is also obtained from image patch embeddings extracted using a frozen vision encoder. Subsequently, following the principles of deep feature separation Bousmalis et al. (2016), we employ two linear layers $\Phi_{\mathcal{V}}$ and $\Phi_{\mathcal{T}}$ to decouple $\mathbf{z}_{\mathcal{P}}$ into two components $\mathbf{z}_{\mathcal{P}}^{\mathcal{V}}$ and $\mathbf{z}_{\mathcal{P}}^{\mathcal{T}}$ in the visual and textual domains.

$$\mathbf{z}_{\mathcal{P}}^{\mathcal{V}} = \Phi_{\mathcal{V}}(\mathbf{z}_{\mathcal{P}}), \quad \mathbf{z}_{\mathcal{P}}^{\mathcal{T}} = \mathrm{FC}_{\mathcal{T}}(\mathbf{z}_{\mathcal{P}}) \tag{5}$$

To ensure that $\mathbf{z}_{\mathcal{P}}^{\mathcal{V}}$ and $\mathbf{z}_{\mathcal{P}}^{\mathcal{T}}$ capture distinct domain information, we utilize an orthogonal loss Bousmalis et al. (2016); Dong et al. (2024) for constraint. The orthogonal loss $\mathcal{L}_{\mathrm{ortho}}$ is defined as follows:

$$\mathcal{L}_{\mathrm{ortho}} = \langle \mathbf{z}_{\mathcal{P}}^{\mathcal{V}}, \mathbf{z}_{\mathcal{P}}^{\mathcal{T}^{\top}} \rangle^2 + \langle \mathbf{z}_{\mathcal{P}}^{\mathcal{T}}, \mathbf{z}_{\mathcal{P}}^{\mathcal{V}^{\top}} \rangle^2 \tag{6}$$

Additionally, to ensure that the two features separated by orthogonal decomposition represent meaningful embeddings, we introduce two constraints: (a) A contrastive learning constraint $\mathcal{L}_{\text{contra}}$ is applied to encourage proximity between $\mathbf{z}_{\mathcal{P}}^{\mathcal{V}}$ and $\mathbf{z}_{\mathcal{P}}^{\mathcal{T}}$ within the same batch, enforcing them as mappings of the same semantics across different domains; (b) $\mathbf{z}_{\mathcal{P}}^{\mathcal{V}}$ is constrained by the visual domain output of SDI I, aligning with the triplet-based inference paradigm. Simultaneously, $\mathbf{z}_{\mathcal{P}}^{\mathcal{T}}$ is constrained through the MSR module together with the textual domain representation $\mathbf{z}_{\mathcal{T}}$. The above constraints will be detailed in Section 3.5.

**Mutual Shift Restriction (MSR).** The MSR module focuses on capturing the semantic shift between reference and target embeddings, aligning it with the modified text embeddings via contrastive learning. Using the reference visual tokens $\mathbf{z}_{\mathcal{V}}$ and decomposed pseudo tokens $\mathbf{z}_{\mathcal{P}}^{\mathcal{T}}$ from the textual domain, MSR employs multi-head self-attention (SA) to refine and emphasize their semantic differences. To achieve this, a dual-path design mutually learns the semantic shift in both reference-to-target and target-to-reference directions. The mutual shift modeling process from reference to target and target to reference is represented as follows.

$$\mathbf{z}_{\delta,\text{ref}}^{(i)} = \mathcal{F}_{\text{SA}}(Q = \mathbf{z}_{\mathcal{V}}, K = \mathbf{z}_{\delta,\text{ref}}^{(i-1)}, V = \mathbf{z}_{\delta,\text{ref}}^{(i-1)}) \tag{7a}$$

$$\mathbf{z}_{\delta,\text{tar}}^{(i)} = \mathcal{F}_{\text{SA}}(Q = \mathbf{z}_{\mathcal{P}}^{\mathcal{T}}, K = \mathbf{z}_{\delta,\text{tar}}^{(i-1)}, V = \mathbf{z}_{\delta,\text{tar}}^{(i-1)}) \tag{7b}$$

where $i$ refers to the SA layer index, $\mathbf{z}_{\delta,\text{ref}}^{(0)} = \mathbf{z}_{\mathcal{P}}^{\mathcal{T}}$, and $\mathbf{z}_{\delta,\text{tar}}^{(0)} = \mathbf{z}_{\mathcal{V}}$. This iterative process enables the MSR module to refine the embeddings continuously by concentrating on the interaction between reference and target embeddings. By alternating query roles, the module effectively isolates their semantic differences. To extract the final shift semantics, we utilize tokens from the output of the last attention layer (denoted as $-1$), and the final mutual shift semantics representation is obtained by averaging embeddings: $\mathbf{z}_{\delta} = (\mathbf{z}_{\delta,\text{ref}}^{(-1)} + \mathbf{z}_{\delta,\text{tar}}^{(-1)})/2$.

## 3.5 OPTIMIZATION AND INFERENCE

**Training.** During the training of PANDA, we focus on decomposing the pseudo domain of the synthetic images and aligning it separately with the visual and textual domains through three constraints: (1) $\mathcal{L}_{\text{OSD}}$, which includes the orthogonal loss $\mathcal{L}_{\text{ortho}}$ to decouple $\mathbf{z}_{\mathcal{P}}$ and the contrastive loss $\mathcal{L}_{\text{proxi}}$ to maintain the semantic proximity between the two components $\mathbf{z}_{\mathcal{P}}^{\mathcal{V}}$ and $\mathbf{z}_{\mathcal{P}}^{\mathcal{T}}$; (2) $\mathcal{L}_{\mathcal{V}}$, a contrastive loss aligning $\mathbf{z}_{\mathcal{P}}^{\mathcal{V}}$ with the multimodal semantics $\mathbf{z}_{\mathcal{V}}$ from the reference and modified text, simulating the inference paradigm; (3) $\mathcal{L}_{\mathcal{T}}$, which employs the MSR module to constrain the mutual shift semantics $\mathbf{z}_{\delta}$ and the modified text semantic $\mathbf{z}_{\mathcal{T}}$ through a contrastive loss. Specifically, the contrastive loss we utilize is a Batch-Based Classification (BBC) loss commonly employed in the CIR task Saito et al. (2023); Wen et al. (2024); Chen et al. (2024a).

$$\mathcal{L}_{\text{BBC}}(\mathbf{z}_{\text{query}}, \mathbf{z}_{\text{tar}}) = \frac{1}{B} \sum_{i=1}^{B} -\log \frac{\exp \kappa(\mathbf{z}_{\text{query}}^{i}, \mathbf{z}_{\text{tar}}^{i})}{\sum_{j=1}^{B} \exp \kappa(\mathbf{z}_{\text{query}}^{i}, \mathbf{z}_{\text{tar}}^{j})} \tag{8}$$

where $B$ represents the batch size, the kernel $\kappa()$ is the inner product resulting in cosine similarity. $\mathbf{z}_{\text{query}}$ denotes the query-side representation, and $\mathbf{z}_{\text{tar}}$ signifies the target-side representation.

Our overall loss $\mathcal{L}_{\text{overall}}$ can be expressed as follow, where $\lambda$ is the trade-off hyper-parameter:

$$\begin{cases} \mathcal{L}_{\mathcal{V}} = \mathcal{L}_{\text{BBC}}(\mathbf{z}_{\mathcal{M}}, \mathbf{z}_{\mathcal{P}}^{\mathcal{V}}) \\ \mathcal{L}_{\mathcal{T}} = \mathcal{L}_{\text{BBC}}(\mathbf{z}_{\delta}, \mathbf{z}_{\mathcal{T}}) \\ \mathcal{L}_{\text{OSD}} = \mathcal{L}_{\text{ortho}} + \mathcal{L}_{\text{BBC}}(\mathbf{z}_{\mathcal{P}}^{\mathcal{V}}, \mathbf{z}_{\mathcal{P}}^{\mathcal{T}}) \\ \mathcal{L}_{\text{overall}} = \mathcal{L}_{\mathcal{V}} + \lambda(\mathcal{L}_{\mathcal{T}} + \mathcal{L}_{\text{OSD}}) \end{cases} \tag{9}$$

**Inference.** During the inference phase, as illustrated in Figure 2, for the real triplet $\{(I_{\text{ref}}^{*}, T_{\text{mod}}^{*}, I_{\text{tar}}^{*})\}$, we employ SDI I to obtain the query-side representation $\mathbf{z}_{\text{query}}^{*}$ and SDI III to extract the gallery-side representation $\mathbf{z}_{\text{gallery}}^{*}$. The similarity between these representations is assessed using inner products, followed by ranking based on the computed similarity scores.

## 4 EXPERIMENT

We present a detailed demonstration of our experimental setting in Section 4.1, report the results of our evaluations in Section 4.2, and provide comprehensive analyses in Section 4.3.

### 4.1 EXPERIMENTAL SETTING.

**Datasets.** To facilitate a fair performance comparison, we adhere strictly to the testing setups established in prior studies Saito et al. (2023); Lin et al. (2024); Baldrati et al. (2023) across all datasets. **(i) CIRR** Liu et al. (2021) comprises approximately 21K open-domain images sourced from the NLVR2 dataset Suhr et al. (2019). To reduce false negatives, annotations ensure the modification text applies to a single image pair, excluding any relevance to other pairs sharing the same reference image. We evaluate our approach on the CIRR test set, consisting of 4.1K triplets. **(ii) CIRCO** Baldrati et al. (2023), derived from the COCO Lin et al. (2014) dataset, addresses false negatives more comprehensively. Unlike other datasets, each CIRCO sample includes a reference image, a modification text, and multiple target images. Our evaluation uses the CIRCO test set, consisting of 800 samples. **(iii) FashionIQ** Wu et al. (2021) focuses on fashion items from three categories: Dresses, Shirts, and Tops&Tees. In line with prior studies, we use the validation set for evaluation. **(iv) Shoes** Guo et al. (2018) is an e-commerce dataset with 4,658 validation queries, following the split used in previous work Guo et al. (2018).

**Evaluation Metrics.** For **(i) CIRR**, as suggested by prior work Saito et al. (2023); Jiang et al. (2024a), we use a combination of evaluation criteria, including $R@K$, $R_{subset}@K$, and the average of R@5 and $R_{subset}@1$. Notably, $R_{subset}@K$ restricts candidate target images to those semantically similar to the correct target image, addressing the issue of false negatives. For **(ii) CIRCO**, due to the presence of multiple ground truths, we follow previous work Baldrati et al. (2023); Lin et al. (2024) and adopt Average Precision (mAP) as a more fine-grained metric. For the **(iii) FashionIQ and Shoes** datasets, in line with previous studies Lin et al. (2024); Chen et al. (2024a), we employ recall at rank $K$ (R@$K$) as the evaluation metric, specifically adopting R@10 and R@50.

**Implementation Details.** We use Stable Diffusion v3 Esser et al. (2024) as the pseudo target image generator, producing 5 images at 512×512 resolution per caption using 20 sampling steps. We randomly sample image subsets of specified sizes from the CC3M dataset Sharma et al. (2018) to construct pseudo triplets. We train the model using up to 100K pseudo-samples. Vicuna-13B-V0.2 Chiang et al. (2023) serves as the LLM. Following the BLIP-2 design Li et al. (2023a), we initialize the encoders using the its pretrained model with ViT-L Radford et al. (2021), and optimize with AdamW Loshchilov & Hutter (2019) using a batch size of 64, an initial learning rate of 1e-5, and a cosine annealing schedule over 50 epochs. All model training and inference are performed on a V100 GPU. All methods utilize ViT-L as the visual backbone.

Table 1: Results on the CIRR dataset Liu et al. (2021). The best and second-best results are highlighted in bold and underlined, respectively. Avg stands for the average of R@5 and $R_{subset}@1$.

| Method | R@$K$ | | | $R_{subset}@K$ | | | Avg |
|---|---|---|---|---|---|---|---|
| | $K$=1 | $K$=5 | $K$=10 | $K$=1 | $K$=2 | $K$=3 | |
| Pic2word Saito et al. (CVPR'23) | 23.90 | 51.70 | 65.30 | 53.28 | 74.10 | 86.27 | 52.49 |
| SEARLE Baldrati et al. (ICCV'23) | 24.87 | 52.31 | 66.29 | 53.80 | 74.31 | 86.94 | 53.06 |
| Context-I2W Tang et al. (AAAI'24) | 25.60 | 55.10 | 68.50 | 58.12 | 78.42 | 88.79 | 56.61 |
| LinCIR Lin et al. (CVPR'24) | 25.04 | 53.25 | 66.68 | 57.11 | 77.37 | 88.89 | 55.18 |
| KEDs Suo et al. (CVPR'24) | 26.40 | 54.80 | 67.20 | 58.16 | 77.91 | 89.23 | 56.48 |
| CIReVL Karthik et al. (ICLR'24) | 24.55 | 52.31 | 64.92 | 59.54 | 79.88 | 89.69 | 55.93 |
| LDRE Yang et al. (SIGIR'24) | 26.53 | 55.57 | 67.54 | 60.43 | 80.31 | 89.90 | 58.00 |
| FTI4CIR Lin et al. (SIGIR'24) | 25.90 | 55.61 | 67.66 | 55.21 | 75.88 | 87.78 | 55.41 |
| ISA Du et al. (ICLR'24) | 30.84 | 61.06 | 73.57 | 64.17 | 80.43 | 89.11 | 62.62 |
| **PANDA (ours)** | **34.11** | **64.55** | **75.94** | **69.48** | **85.98** | **93.16** | **67.02** |

## 4.2 RESULTS

**Quantitative Analysis.** Tables 1, 2, and 3 present the performance results of our PANDA approach compared to existing methods on the CIRR, CIRCO&Shoes, and FashionIQ datasets. Three key observations can be made: (i) Despite the domain differences and varying construction across the four benchmarks, PANDA achieves state-of-the-art performance on these datasets, including general domain CIRR and CIRCO, as well as e-commerce domain Shoes and FashionIQ; (ii) To address the target ambiguity inherent in the CIR task, our Synthetic Image-Oriented training paradigm naturally introduces multiple synthetic images with the same semantics. This leads to significant performance improvements on the $R_{subset}$ metric, specifically designed to mitigate false negatives caused by target ambiguity. Liu et al. (2021); (iii) Although existing methods Lin et al. (2024); Du et al. (2024) propose semantically enriched modeling of individual image tokens and demonstrate some effectiveness, their performance is limited by the lack of semantic interaction and alignment within a triplet structure. In contrast, our approach more effectively captures the core semantics of the modified text, resulting in more precise and comprehensive fulfillment of modification requirements.

Table 2: Results on the CIRCO Baldrati et al. (2023) and Shoes Guo et al. (2018) datasets. The best and second-best results are highlighted in bold and underlined, respectively.

| Method | CIRCO 2023 | | | | Shoes 2018 | |
|---|---|---|---|---|---|---|
| | mAP@5 | mAP@10 | mAP@25 | mAP@50 | R@10 | R@50 |
| Image + Text | 4.32 | 5.24 | 6.49 | 7.07 | 13.11 | 30.76 |
| Captioning | 8.33 | 8.98 | 10.17 | 10.75 | 16.06 | 32.78 |
| Pic2word (CVPR'23) | 8.72 | 9.51 | 10.46 | 11.29 | 22.34 | 46.17 |
| SEARLE (ICCV'23) | 11.68 | 12.73 | 12.73 | 14.33 | 23.51 | 47.64 |
| LinCIR (CVPR'24) | 12.59 | 13.58 | 15.00 | 15.85 | 24.23 | 48.99 |
| ISA (ICLR'24) | 11.33 | 12.25 | 13.42 | 13.97 | 28.73 | 53.89 |
| FTI4CIR (SIGIR'24) | 15.05 | 16.32 | 18.06 | 19.05 | 29.21 | 55.40 |
| **PANDA (ours)** | **16.59** | **17.84** | **19.82** | **20.59** | **31.97** | **58.38** |

Table 3: Results on the FashionIQ dataset Wu et al. (2021). The best and second-best results are highlighted in bold and underlined, respectively.

| Method | Dresses | | Shirts | | Tops&Tees | | Avg |
|---|---|---|---|---|---|---|---|
| | R@10 | R@50 | R@10 | R@50 | R@10 | R@50 | |
| Pic2word (CVPR'23) | 20.00 | 40.20 | 26.20 | 43.60 | 27.90 | 47.40 | 34.20 |
| SEARLE (ICCV'23) | 21.57 | 44.47 | 30.37 | 47.49 | 30.90 | 51.76 | 37.76 |
| LinCIR (CVPR'24) | 20.92 | 42.44 | 29.10 | 46.81 | 28.81 | 50.18 | 36.39 |
| KEDs (CVPR'24) | 21.70 | 43.80 | 28.90 | 48.00 | 29.90 | 51.90 | 37.35 |
| CIReVL (ICLR'24) | 24.79 | 44.76 | 29.49 | 47.40 | 31.36 | 53.65 | 38.56 |
| LDRE (SIGIR'24) | 22.93 | 46.76 | 31.04 | 51.22 | 31.57 | 53.64 | 39.53 |
| Context-I2W (AAAI'24) | 23.10 | 45.30 | 29.70 | 48.60 | 30.60 | 52.90 | 38.35 |
| ISA (ICLR'24) | 25.48 | 45.51 | 29.64 | 48.68 | 32.94 | 54.31 | 39.43 |
| FTI4CIR (SIGIR'24) | 24.39 | 47.84 | 31.35 | 50.59 | 32.43 | 54.21 | 40.14 |
| **PANDA (ours)** | **25.88** | **49.78** | **31.45** | **51.62** | **33.30** | **57.68** | **41.62** |

**Qualitative Analyses.** Our approach is visualized on representative datasets from general and e-commerce domains, CIRR and FashionIQ, in comparison to the SOTA method Lin et al. (2024). As shown in Figure 3, our model handles complex, fine-grained modifications and generalizes well when multiple targets meet the requirements (e.g., white Mickey T-shirt).

### 4.3 ABLATION STUDIES

**Effects of Different Components.** Table 4 provides an ablation study to validate the contribution of each key component, followed by the detailed analysis below: (i)) For the loss $\mathcal{L}_{\mathcal{V}}$, we modify

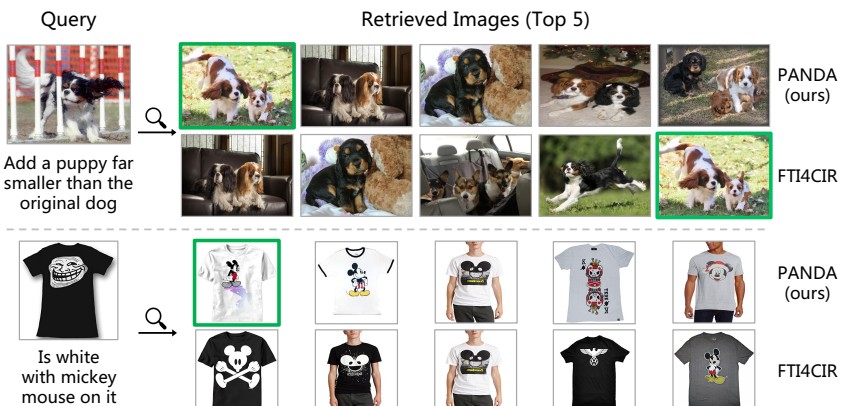

Figure 3: Qualitative results on general and e-commerce domains, with green-boxed ground truths.

$\mathcal{L}_{\mathrm{BBC}}(\mathbf{z}_{\mathcal{M}}, \mathbf{z}_{\mathcal{P}}^{\mathcal{V}})$ to $\mathcal{L}_{\mathrm{BBC}}(\mathbf{z}_{\mathcal{M}}, \mathbf{z}_{\mathcal{P}})$, which leads to over-fitting to the pseudo domain of the synthetic images, resulting in the loss of modeling the semantic shift between the reference and target described by the modified text in the triplet. (ii) For the $\mathcal{L}_{\mathcal{T}}$ term, its removal hinders the modeling of the semantic shift between the reference and target described by the modified text in the triplet. (iii) We remove the loss $\mathcal{L}_{\mathrm{OSD}}$, which eliminates the correlation constraint between $\mathbf{z}_{\mathcal{P}}^{\mathcal{V}}$ and $\mathbf{z}_{\mathcal{P}}^{\mathcal{T}}$, making it difficult to enforce alignment within the visual and textual domains. (iv) For the semantic constraint design of $\mathbf{z}_{\mathcal{P}}^{\mathcal{T}}$ in the textual domain, we consider a naive constraint method, Mod2Tar, where the modified text directly serves as the constraint. We observe that this setup yields some improvements in datasets where the modified text plays a dominant role Baldrati et al. (2023). However, it also leads to cases where the modified text dominates the retrieval results, ignoring the reference and thus impacting performance. (v) We replace SDI's semantic interaction mechanism with the existing Concat method, which concatenates visual features and text embeddings. This method lacks cross-modal interaction, highlighting the necessity of the SDI module, which provides a solid embedding foundation for subsequent OSD and MSR modules to impose constraints.

Table 4: Ablation study on different components of PANDA.

| Method | CIRR | CIRCO |
|---|---|---|
| w/o $\mathcal{L}_{\mathcal{V}}$ | 63.62 | 12.94 |
| w/o $\mathcal{L}_{\mathcal{T}}$ | 65.78 | 17.43 |
| w/o $\mathcal{L}_{\mathrm{OSD}}$ | 64.18 | 15.18 |
| w/o MSR | 66.70 | 16.59 |
| w/o SDI | 60.30 | 15.28 |
| **PANDA** | **67.02** | **18.71** |

Table 5: Ablation of data scales in the CIRR dataset.

| Methods | Scale | Avg |
|---|---|---|
| Pic2word | 3M | 52.49 |
| ISA | 3M | 62.62 |
| FTI4CIR | 100K | 55.41 |
| SIO-1K | 1K | 57.19 |
| SIO-5K | 5K | 63.46 |
| SIO-10K | 10K | **67.02** |

Table 6: Synthetic images $N_{\mathrm{gen}}$ per caption.

| Setting | Avg |
|---|---|
| $N_{\mathrm{gen}} = 1$ | 64.28 |
| $N_{\mathrm{gen}} = 2$ | 65.47 |
| $N_{\mathrm{gen}} = 3$ | 66.45 |
| $\mathbf{N_{gen} = 5}$ | 67.02 |
| $N_{\mathrm{gen}} = 7$ | 66.94 |
| $N_{\mathrm{gen}} = 10$ | 66.83 |

**Data Scales.** We compare the performance on the CIRR dataset under different training data scales. We randomly sample image subsets of specified sizes from the CC3M dataset to construct pseudo triplets. As shown in Table 5, due to the similarity between pseudo triplets and the inference-phase paradigm, our approach outperforms existing methods with 1-2 orders of magnitude less data.

**Number of Synthetic Images per Caption.** We evaluate the impact of the number of pseudo target images generated per caption, as shown in Table 6. Increasing $N_{\mathrm{gen}}$ improves generalization by aligning with target ambiguity in the CIR task, enhancing performance. However, excessively high $N_{\mathrm{gen}}$ reduces retrieval accuracy, indicating that an appropriate $N_{\mathrm{gen}}$ represents a trade-off.

**Autophagy Phenomenon.** We observe an Autophagy Phenomenon where performance decreases as pseudo dataset size increases. However, after adding our decoupling method $\mathcal{L}_{\mathrm{OSD}}$, this issue is resolved, as shown by the average metrics on the CIRCO dataset in Table 7.

Table 7: Ablation of the Autophagy.

| Methods | Scale | w/ $\mathcal{L}_{\text{OSD}}$ | w/o $\mathcal{L}_{\text{OSD}}$ |
|---|---|---|---|
| SIO-10K | 10K | 17.76 | 16.82 |
| SIO-30K | 30K | 18.03 | 16.07 |
| SIO-50K | 50K | 18.22 | 15.68 |
| SIO-100K | 100K | **18.71** | 15.18 |

Table 8: Ablation of different LLMs.

| Model | CIReVL | PANDA |
|---|---|---|
| w/o LLMs | 10.70 | 16.90 |
| LLAMA2-13B | 11.04 | 17.98 |
| Vicuna-13B | 13.88 | **18.71** |
| LLAMA2-70B | 11.25 | 18.26 |

**Different LLMs.** In SIO, LLMs play a role in adding or replacing specific objects. As shown in Table 8, our approach demonstrates robustness across different LLMs (including LLAMA2 Touvron et al. (2023) and Vicuna Chiang et al. (2023)). Notably, we also employ a method without LLMs, replacing detected objects with random categories from ImageNet1K Russakovsky et al. (2015) for the image generation model, yielding competitive results. While recent training-free methods for ZS-CIR heavily rely on LLMs for summarizing reference captions and modified text, our approach outperforms the representative CIReVL Karthik et al. (2024a) without depending on LLM performance, as illustrated in Table 8 for the average metric on the CIRCO dataset.

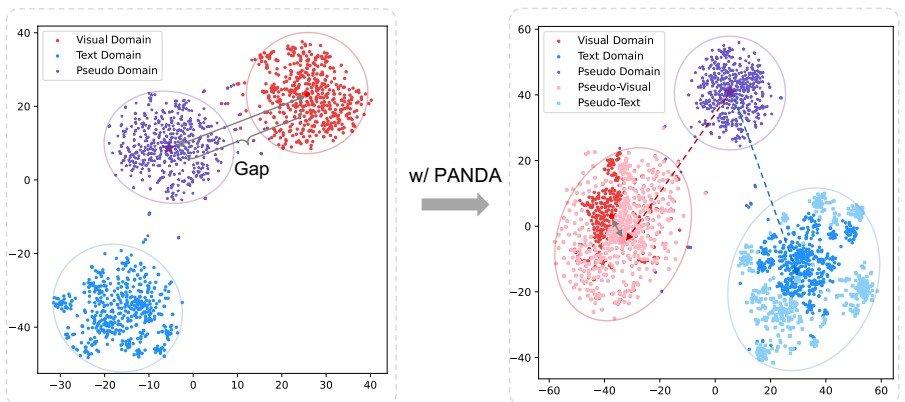

(a) Pseudo and Visual Domain Gap w/o PANDA.   (b) Pseudo and Visual Domain Gap w/ PANDA.

Figure 4: t-SNE visualization of decoupled Pseudo Domain using the PANDA approach.

**Domain Gap.** We visualize the embedding distributions of the reference real images, modified text, pseudo target images, and decoupled embeddings $\mathbf{z}_{\mathcal{V}}$ and $\mathbf{z}_{\mathcal{P}}^{\mathcal{T}}$ in the pseudo triplet using t-SNE. As shown in Figure 4, our approach significantly reduces the domain gap between the decoupled $\mathbf{z}_{\mathcal{V}}$ and $\mathbf{z}_{\mathcal{P}}^{\mathcal{T}}$ in the visual and textual domains, facilitating semantic optimization.

## 5 CONCLUSION

In this paper, we offer the insight that current ZS-CIR training methods lack explicit semantic learning for triplets, limiting their capacity for fine-grained or multi-attribute modifications. To address this, we introduce the Synthetic Image-Oriented training paradigm, leveraging synthetic images to swiftly form pseudo triplets while addressing target ambiguity in CIR. Additionally, to mitigate overfitting caused by pseudo images, we propose the Pseudo domAiN Decoupling-Alignment (PANDA) model, which decouples the pseudo domain and applies separate alignment constraints. Comprehensive experiments demonstrate the effectiveness of our proposed training paradigm and approach.

**Limitations.** Although the proposed Synthetic Image-Oriented training paradigm allows for the quick construction of pseudo-triplets, enabling the model to efficiently learn the correspondence between triplet components, the modified text in real-world scenarios may involve more complex semantics, such as comparatives or multiple conjunctions. Our next research goal is to leverage synthetic images' inherent target ambiguity to adapt to these more complex semantic cases.

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

## 6 APPENDIX

### 6.1 MORE ABLATION STUDIES.

**Different Image Generation Models.** To assess the robustness of our proposed Synthetic Image-Oriented (SIO) training paradigm, we evaluate the impact of different generation models—SDXL Turbo Sauer et al. (2023), Stable Diffusion v2 Rombach et al. (2022), Stable Diffusion v3 Esser et al. (2024), and Stable Diffusion XL Podell et al. (2023)—on pseudo target image generation under the same caption, as presented in Table 9. Notably, our proposed SIO paradigm achieves consistent performance regardless of using high-performance, high-resolution models or faster generation models (207 ms per image). We attribute this to the fact that our approach does not rely on pixel-level information of the synthetic images but instead leverages the OSD model to map the semantic embeddings of the synthetic images across domains.

**Image Editing Methods.** As the image editing task is closely related to composed image retrieval, we explore generating target images using representative image editing models (InsPix2Pix Brooks et al. (2023), SmartEdit Huang et al. (2024) and MGIE Fu et al. (2024)) when constructing pseudo triplets, as shown in Table 10. We observe a significant performance drop compared to generating images based on captions. This decline is likely due to the fundamental difference between the two tasks: image editing typically modifies only specific objects while keeping the rest unchanged, whereas composed image retrieval imposes less stringent constraints.

Table 9: Ablation of image generation models for pseudo target image generation.

| Model | CIRR | CIRCO |
|---|---|---|
| SDXL Turbo | 65.19 | 17.42 |
| Stable Diffusion v2 | 66.24 | 17.98 |
| **Stable Diffusion v3** | 67.02 | 18.71 |
| Stable Diffusion XL | 66.56 | 18.34 |

Table 10: Ablation of image editing methods for pseudo data.

| Model | CIRR | CIRCO |
|---|---|---|
| InsPix2Pix | 60.05 | 10.34 |
| SmartEdit | 61.38 | 11.26 |
| MGIE | 61.26 | 10.87 |
| **SIO (ours)** | **67.02** | **18.71** |

**Different pseudo triplet construction methods.** Recent work Jiang et al. (2024b) has approached pseudo triplet generation by using LLMs to describe the differences between captions of two specified images. However, this method significantly relies on the LLM's ability to analyze and infer differences between captions. We compare the performance of both construction paradigms at the same data scale (10K) on the CIRCO dataset, as shown in Table 11. Our proposed SIO paradigm demonstrates greater robustness and superior performance compared to the LLM-dependent method.

Table 11: Results on the CIRCO Baldrati et al. (2023) and Shoes Guo et al. (2018) datasets. The best and second-best results are highlighted in bold and underlined, respectively.

| Model | LLAMA2-13B | LLAMA2-70B | Vicuna-13B |
|---|---|---|---|
| HyCIR | 10.26 | 12.13 | 14.29 |
| PANDA (ours) | 17.98 | 18.26 | 18.71 |

**Trade-off between $w_{\text{fine}}$ and $w_{\text{coarse}}$.** We conduct an ablation study on the ratio between $w_{\text{fine}}$ and $w_{\text{coarse}}$, as shown in Figure 5. A larger $w_{\text{fine}}$ value (8:2) facilitates the model's learning of fine-grained semantics among triplets. However, excessive reliance on generated images corresponding to $w_{\text{fine}}$ leads to increased fitting difficulty, resulting in performance degradation.

### 6.2 MORE DETAILED THEORETICAL INSIGHT

In the context of existing ZS-CIR methods, only one target image is paired with a reference and modified text, defining a single triplet $x_0 = (I_{\text{ref}}, T_{\text{mod}}, I_{\text{tar}})$ as the root of $f - \mathcal{F}$. Therefore, a linear approximation is achieved:

$$f(x) - \mathcal{F}(x) = k(x - x_0) \tag{10}$$

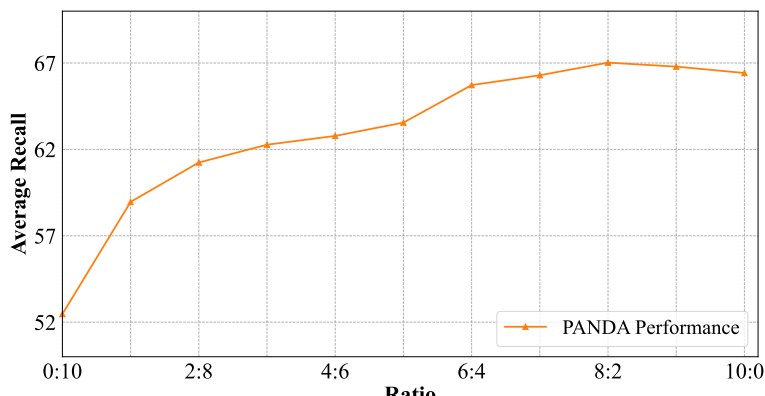

Figure 5: Ablation study of different $w_{\text{fine}}$ and $w_{\text{coarse}}$.

where $k$ is a constant. On the other hand, our approach introduces multiple synthetic target images in a pseudo-triplet as a set of roots $\{x_i = (I_i, T_{\text{mod}}^{\text{fine}}, I_i^{\text{gen}})\}$, leading to a polynomial approximation:

$$f(x) - \mathcal{F}(x) = k\Pi_i(x - x_i) \tag{11}$$

The analyses based on the Weierstrass approximation theorem highlight the potential of our approach to facilitate more complex and accurate approximations of the underlying ground truth mapping function.

**Proof of Weierstrass Approximation Theorem.**    Without loss of generality, let $\mathcal{F}$ be a continuous function on the interval $[0, 1]$, consider the following polynomial series:

$$B_n(\mathcal{F})(x) = \sum_{v=0}^{n} \mathcal{F}(\frac{v}{n})b_{v,n}(x) \tag{12}$$

where $b_{v,n} = \binom{n}{v}x^v(1-x)^{n-v}$ denotes the Bernstein basis polynomials, and $\binom{n}{v}$ is a binomial coefficient. According to the properties of the Bernstein basis polynomials, we have:

$$B_n(\mathcal{F})(x) - \mathcal{F}(x) = \sum_v [\mathcal{F}(\frac{v}{n}) - \mathcal{F}(x)]b_{v,n}(x) \tag{13}$$

so that

$$|B_n(\mathcal{F})(x) - \mathcal{F}(x)| \leq \sum_v |\mathcal{F}(\frac{v}{n}) - \mathcal{F}(x)|b_{v,n}(x) \tag{14}$$

Since $\mathcal{F}$ is uniformly continuous, given $\varepsilon > 0$, there exists $\delta > 0$ such that $|\mathcal{F}(a) - \mathcal{F}(b)| < \varepsilon$ for any $|a - b| < \delta$, then according to Chebyshev's Inequality, we have:

$$\sum_{|x-k/n|\geq\delta} b_{v,n}(x) \leq \sum_v \delta^{-2}\big(x - \frac{v}{n}\big)b_{v,n}(x) = \delta^{-2}\frac{x(1-x)}{2} < \frac{1}{4}\delta^{-2}n^{-1} \tag{15}$$

which leads to

$$\lim_{n \to \infty} B_n(\mathcal{F}) = \mathcal{F} \tag{16}$$

holds uniformly on the interval $[0, 1]$, which satisfies approximating $\mathcal{F}$ with polynomial functions and gives the proof of the Weierstrass approximation theorem.    $\square$