# OpenReview forum: "Pseudo Meets Zero: Boosting Zero-Shot Composed Image Retrieval with Synthetic Images"
_ICLR.cc/2025/Conference — Submitted to ICLR 2025_

### Official Review · Reviewer_exLu · 2024-10-31

**Soundness:** 3
**Presentation:** 2
**Contribution:** 2
**Rating:** 5
**Confidence:** 4

**Summary:**

This paper presents a zero-shot composed image retrieval (ZS-CIR) method called PANDA (Pseudo domAiN Decoupling-Alignment). PANDA leverages synthetic images through an automated pipeline for pseudo-target image generation, enabling efficient triplet construction for ZS-CIR model training. Observing that synthetic images lie between visual and textual domains, the paper proposes an Orthogonal Semantic Decoupling module to disentangle this pseudo domain. With additional constraints, PANDA achieves state-of-the-art results on CIR benchmarks.

**Strengths:**

1. This paper proposes a training approach that successfully leverages synthetic pseudo-target images for CIR triplet construction.

2. An investigation is conducted using the Weierstrass Approximation Theorem on synthetic target images that lie between the visual and textual domains.

3. Various solutions are proposed, including Shared Domain Interaction, Orthogonal Semantics Decoupling, and Mutual Shift Restriction, to address issues related to the Autophagy phenomenon.

4. Extensive experiments and ablation studies are conducted with accompanying theoretical analysis.

**Weaknesses:**

1. Several recent references on zero-shot composed image retrieval are missing, which demonstrate superior performance compared to PANDA.

[1] Geonmo Gu et al., CompoDiff: Versatile Composed Image Retrieval With Latent Diffusion, TMLR
[2] YK Jang et al., Spherical Linear Interpolation and Text-Anchoring for Zero-shot Composed Image Retrieval, ECCV 2024
[3] Kai Zhang et al., MagicLens: Self-Supervised Image Retrieval with Open-Ended Instructions, ICML 2024

2. Compared to recent methods, PANDA’s performance falls short despite the model's complexity in training for composed image retrieval.

3. Each component in PANDA’s training pipeline appears overly complex, with minimal performance improvement observed in Table 4.

**Questions:**

Has there been any investigation into training PANDA with larger-scale datasets? The current model seems to be trained with only 100K pairs, and it would be useful to assess whether PANDA’s performance could improve with additional samples. Although Table 5 presents some results regarding dataset scales, it remains unclear if further scaling could yield performance comparable to recent works. This is particularly relevant given the use of multiple generative models, such as Stable Diffusion v3 and Vicuna, in the training pipeline. As the number of pseudo triplets increases, the likelihood of introducing noisy (hallucinated) samples may also increase.

---

### Official Review · Reviewer_zdpD · 2024-11-02

**Soundness:** 3
**Presentation:** 3
**Contribution:** 3
**Rating:** 3
**Confidence:** 4

**Summary:**

This paper advances Zero-Shot Composed Image Retrieval (ZS-CIR) by introducing a synthetic image-based training paradigm coupled with a Pseudo domAiN Decoupling-Alignment (PANDA) model for effective feature handling. The approach achieves competitive performance while reducing training data requirements.

**Strengths:**

1. The paper collects a synthetic dataset which is good.
2. The experiment of Autophagy Phenomenon is interesting and more explanation is expected.

**Weaknesses:**

Weakness:

1. Lack of compared method: Tables 1-3 only include zero-shot methods trained on text-image pairs, excluding those using synthetic triplets. The reported improvements may primarily result from additional synthetic training data rather than architectural innovation. A crucial baseline comparison with TransAgg, a zero-shot method that also leverages synthetic data, is missing from the evaluation.
2. Lack of comparison with simply fine-tuning: is it possible to directly fine-tune existing models on the synthetic dataset? So that it is clear the benefit from datasets and the proposed architecture.
3. Readability: the paper uses both SDI (I, II, III) and (M, V, T) to refer to multimodal, visual, and text processing， which is redundant. Additionally, Figure 2 contains a discrepancy where the target image is processed by SDI III (designated for text) during inference, contradicting the caption. These inconsistencies impair the paper's readability.
4. One of the paper’s main contribution is a synthetic dataset, but visualization of  synthetic images is missing.

[1]Zero-shot Composed Text-Image Retrieval, Yikun Liu, Jiangchao Yao, Ya Zhang, Yanfeng Wang, Weidi Xie, BMVC 2023.

**Questions:**

Question:

1. The paper's use of the Weierstrass Approximation Theorem lacks proper justification. The claimed causal relationship between multiple synthetic targets and higher-degree polynomial functions appears not convincing, particularly given the fixed model architecture.
2. More illustration is needed for eq.6, is <> calculating cosine similarity?
3. interesting experiment for the Autophagy Phenomenon， can you share insights behind it? Is it because the generated SIO dataset is of bad quality and how does Losd solve this problem?

**Details Of Ethics Concerns:**

no concern

---

### Official Review · Reviewer_wwUB · 2024-11-04

**Soundness:** 3
**Presentation:** 2
**Contribution:** 2
**Rating:** 5
**Confidence:** 4

**Summary:**

This paper proposes a CIR framework that utilizes synthetically generated pseudo-triplets based on a reference image: conditioning text is generated with an LLM, and the target image is created using a text-to-image generative model. To address overfitting issues associated with using pseudo-triplets, the authors introduce Pseudo domAiN Decoupling-Alignment (PANDA) to mitigate the Autophagy phenomenon. PANDA comprises three key components: the Orthogonal Semantic Decoupling module (OSD), Shared Domain Interaction (SDI), and Mutual Shift Constraint (MSR). The approach demonstrates strong performance across various benchmarks. However, the positioning and comparative analysis of the proposed method relative to existing approaches are somewhat unclear, and additional, more detailed ablation studies with explanations would be beneficial.

**Strengths:**

1. The motivation which mitigate the Autophagy phenomenon (reducing the domain gap between pseudo domain and real image domain) seems solid.
2. The effectiveness of method is demonstrated through excessive experiments and it seems that the desired goal is achieved. Moreover, it achieves strong performance compared to other ZS-CIR methods.
3. The introduction and related work sections are clear and easy to follow.

**Weaknesses:**

1. My primary concern is the paper’s positioning and its comparison with existing CIR methods. Numerous existing methods generate synthetic CIR triplets to enhance performance, such as MagicLens, CompoDiff, and CoVR. These methods also create CIR triplets and use them to boost retrieval results. Therefore, it would be beneficial for this paper to include a performance comparison with these approaches.

Additionally, I question whether the main claim—reducing the domain gap between the pseudo and real image domains—holds across other publicly available datasets. For example, CompoDiff also synthetically generates target images. Testing the effectiveness of PANDA on a subset of the CompoDiff dataset would clarify its generalizability. In other cases, methods like MagicLens and CoVR generate conditioning text with LLMs from similar real images, potentially not suffering from the Autophagy phenomenon. I wonder how PANDA would perform with these models. Although their datasets are quite large, I wonder about the results of models trained with a small portion of dataset.

[1] MagicLens: Self-Supervised Image Retrieval with Open-Ended Instructions, Zhang et al., ICML 2024
[2] CompoDiff: Versatile Composed Image Retrieval With Latent Diffusion, Gu et al., TMLR 2024
[3] CoVR: Learning Composed Video Retrieval from Web Video Captions, Ventura et al., AAAI 2024

2. The proposed method (PANDA) is highly complex, making it challenging to understand the entire mechanism and to identify which components genuinely contribute to its effectiveness. The notations also seem overly detailed.

- What is the main motivation behind orthogonal semantic decoupling (OSD)? It appears to decouple the image and text parts of the pseudo target visual token, but it’s unclear how this contributes to mitigating overfitting to the pseudo domain.
- The rationale for L_T  is also unclear. While the motivation behind Mutual Shift Constraint (MSR) is reasonable, why must the mutual shift semantics representation be constrained by 𝑍_T?
-It seems that the primary model structure resembles the BLIP-2 model. The Shared Domain Interaction (SDI) part, which uses learnable tokens, closely resembles BLIP-2’s Q-former. Clarifying these aspects and analyses on network architecture would improve comprehension.

3. I think more explanation should be incorporated in the ablation study. Given the method's complexity, it is difficult to determine if the ablation studies genuinely demonstrate each component’s effectiveness. I'm not sure it's possible but more fair comparison would be good to add in ablation studies

- In ablation study, the impact of L_V and OSD  appear critical, and, as shown in (9), are related. Removing L_V while retaining OSD and other losses naturally leads to performance degradation. But, the performance difference between removing OSD and L_V is significant and wonder the rationale behind this. Moreover, I wonder the results when L_BBC(Z_M, Z_P) without both OSD and L_V.
- Similarly, L_T and MSR seems closely related. Therefore, more detailed explanations (or additional experiments) are needed to fairly compare these components.

Currently, each ablation study removes a single component in turn. It would be valuable to see the results when individual losses or components are added separately. Ideally, if possible, the paper could include results for various combinations of components.

4. Lastly, an analysis of the pseudo dataset (e.g., fine-grained vs. coarse-grained instances) would be valuable.

**Questions:**

All questions are described in weakness sections.

---

### Official Review · Reviewer_PVEC · 2024-11-04

**Soundness:** 2
**Presentation:** 1
**Contribution:** 2
**Rating:** 3
**Confidence:** 4

**Summary:**

This paper introduces a method that leverages the pre-trained knowledge of diffusion modal and LLM to generate pseudo triplets for training. To address the domain gap of the pseudo target image, this paper introduces PANDA, a BLIP-based architecture with a complex training approach. Extensive experiments demonstrate the proposed approach outperforms existing state-of-the-art methods across four CIR datasets.

**Strengths:**

1. The motivation is easy to understand.
2. It is interesting to propose a module to address the domain gap of pseudo and real images for CIR.
3. Extensive experiments and ablation studies show the efficiency of PANDA.

**Weaknesses:**

1. The setting of this paper is inconsistent with standard ZS-CIR tasks [1,2,3,4,5,6,7], which is unfair to compare. It seems more aligned with what "the Semi-Supervision CIR" [8] aims to address, which generates pseudo-triplets for CIR. This inconsistent setting may cause potential data leakage in the training process, which is a fitting bias of the CIR data, leading to potential data leakage. Moreover, this method required training the entire CLIP model, introducing significant training parameter size, computational resources cost, and time increase, which makes it unfair to compare existing ZS-CIR methods.

2. The novelty is limited. Even though the story of this paper might be hard to understand, the motivation is straightforward, which aims to leverage the pre-trained knowledge of the Diffusion model for generating pseudo-target images and introduce a method for training the CIR model with those pseudo images. However, The paper overlooks some similar existing methods in the CIR domain. For example, Compodiff [9] leverages pre-trained knowledge to generate target images and proposes a pseudo-triplets dataset for CIR. The authors need to acknowledge this prior work and clearly differentiate their method to highlight the proposed method's unique contributions and innovations.

3. The technology contribution is limited. Even the structure of PANDA is complex. The overall design is similar to BLIP-2 [10], which the authors may not compare in their methodology section. Moreover, the learnable tokens also have been explored in the ZS-CIR domain [4]. This method seems to only propose a module of Orthogonal Semantics Decoupling to mitigate over-fitting to the pseudo domain, where the Orthogonal loss comes from existing works. Furthermore, the authors do not explain the reason for decoupling the pseudo domain into visual and textual domains, making it confusing. Additionally, PANDA might face the challenge of generalization as different diffusion models have distinct pseudo domains, requiring PANDA to be re-trained to align with each model.

4. Need more qualitative experiments. This paper provides the domain gap analysis through t-SNE visualization; however, it might not be sufficient. It is necessary to provide more qualitative experiments, such as showing the pseudo-triples the paper generated. One of my main concerns is the efficient and generated data quality of the Fashion domain, which includes fine-grained attribute-relevant details that the Diffusion modal is hard to generate.

5. Insufficient implementation details. Some hyperparameters are not specified (e.g., The hyperparameters of the diffusion model), and the code has not been provided, which impedes the reproducibility and verification of the results.

6. Need more ablation studies. For example, what the influence of the hyper-parameter in Function (4)? What is the generalization when using PANDA for different diffusion modes without re-training? Moreover, CIRR and CIRCO are in some domains, so it is necessary to conduct an ablation study in two different domain datasets (e.g., Fashion-IQ).

Overall, due to the unfair setting with potential data leakage, limited novelty, technology contribution, and insufficient ablation and qualitative experiments. I give a "Reject" recommendation. I will consider raising my score if the authors address my concerns.

Reference

[1] Geonmo Gu, Sanghyuk Chun, Wonjae Kim, Yoohoon Kang, and Sangdoo Yun. Language-only efficient training of zero-shot composed image retrieval. In CVPR, 2024.

[2] Kuniaki Saito, Kihyuk Sohn, Xiang Zhang, Chun-Liang Li, Chen-Yu Lee, Kate Saenko, and Tomas Pfister. Pic2word: Mapping pictures to words for zero-shot composed image retrieval. In CVPR, 2023.

[3] Suo Y, Ma F, Zhu L, et al. Knowledge-enhanced dual-stream zero-shot composed image retrieval[C]//Proceedings of the IEEE/CVF Conference on Computer Vision and Pattern Recognition. 2024: 26951-26962.

[4] Tang Y, Yu J, Gai K, et al. Context-I2W: Mapping Images to Context-dependent Words for Accurate Zero-Shot Composed Image Retrieval[C]//Proceedings of the AAAI Conference on Artificial Intelligence. 2024, 38(6): 5180-5188.

[5] Du Y, Wang M, Zhou W, et al. Image2Sentence based Asymmetrical Zero-shot Composed Image Retrieval[J]. ICLR 2024.

[6] Karthik S, Roth K, Mancini M, et al. Vision-by-language for training-free compositional image retrieval[J]. ICLR 2024.

[7] Alberto Baldrati, Lorenzo Agnolucci, Marco Bertini, and Alberto Del Bimbo. Zero-shot composed image retrieval with textual inversion. In ICCV, 2023.

[8] Jang Y K, Kim D, Meng Z, et al. Visual Delta Generator with Large Multi-modal Models for Semi-supervised Composed Image Retrieval[C]//Proceedings of the IEEE/CVF Conference on Computer Vision and Pattern Recognition. 2024: 16805-16814.

[9] Geonmo Gu, Sanghyuk Chun, HeeJae Jun, Yoohoon Kang, Wonjae Kim, and Sangdoo Yun. Compodiff: Versatile composed image retrieval with latent diffusion. arXiv preprint arXiv:2303.11916, 2023.

[10] Junnan Li, Dongxu Li, Silvio Savarese, and Steven Hoi. Blip-2: Bootstrapping language-image pre-training with frozen image encoders and large language models. In International conference on machine learning, pp. 19730–19742. PMLR, 2023a.

**Questions:**

1. Why do you not compare the difference between CIG and Compodiff?
2. What is the time cost for generating entire pseudo-triplets (including the LLM modification and Diffusion generation stage)?
3. Are you trained in different PANDA for each pseudo-domain of different diffusion models?
4. Is there any selection strategy for pseudo-target images?
5. Could you visualize an example of pseudo triples of the Fashion domain?
6. What are ablation studies on the Fashion-IQ dataset?
7. What is the influence of the hyper-parameter in Function (4)?
8. What is the generalization while using PANDA for difference diffusion modal without re-training
9. There might be a mistake in Figure 2, which does not contain SDI III in the Training phase.

---

> ### Comment · Reviewer_PVEC · 2024-12-02
>
> Unfortunately, the authors have not provided any responses to my previous concerns regarding the setting, novelty, and experimental aspects of the paper. Other reviewers also raised similar concerns. Given the lack of any reply or clarification, I maintain my previous rating and level of confidence.

---

### Meta-Review · Area_Chair_VzkC · 2024-12-08

**Metareview:**

At the initial review stage, all the reviewers have negative opinions.

The concerns are mainly centered around novelty, technical contribution, lack of analysis, lack of comparisons with many related works, and insufficient experiments

As the authors did not provide a rebuttal and the AC agrees with the initial reviews by the reviewers, the AC recommends the rejection of this paper.

**Additional Comments On Reviewer Discussion:**

No rebuttal has been provided.

---

### Decision · Program_Chairs · 2025-01-22

Reject